# Antibacterial, Anti-Biofilm, and Anti-Inflammatory Properties of Gelatin–Chitosan–Moringa-Biopolymer-Based Wound Dressings towards *Staphylococcus aureus* and *Escherichia coli*

**DOI:** 10.3390/ph17050545

**Published:** 2024-04-23

**Authors:** Salma Bessalah, Asim Faraz, Mohamed Dbara, Touhami Khorcheni, Mohamed Hammadi, Daniel Jesuwenu Ajose, Shamsaldeen Ibrahim Saeed

**Affiliations:** 1Livestock and Wildlife Laboratory, Arid Lands Institute (I.R.A.), University of Gabès, Médenine 4119, Tunisia; bessalahsalma@yahoo.fr (S.B.); mohamed.dbara@gmail.com (M.D.); touha2009@gmail.com (T.K.); hedhammadi70@yahoo.fr (M.H.); 2Department of Livestock and Poultry Production, Bahauddin Zakariya University, Multan 60800, Pakistan; drasimfaraz@bzu.edu.pk; 3Department of Microbiology, North-West University, Mmabatho 2735, South Africa; ajosedj@yahoo.com; 4Department of Microbiology, Faculty of Veterinary Science, University of Nyala, Nyala P.O. Box 155, Sudan; 5College of Veterinary Medicine, University of Juba, Central Equatoria, Juba P.O. Box 82, South Sudan; 6Nanotechnology Research Group Faculty of Veterinary Medicine, Universiti Malaysia Kelantan, Kota Bharu 16100, Malaysia

**Keywords:** alternative therapy, biomedical, biopolymers, botanical, natural product, pathogen

## Abstract

In contemporary times, the sustained aspiration of bioengineering and biomedical applications is the progressive advancement of materials characterized by biocompatibility and biodegradability. The investigation of the potential applications of polymers as natural and non-hazardous materials has placed significant emphasis on their physicochemical properties. Thus, this study was designed to investigate the potential of gelatin–chitosan–moringa leaf extract (G–CH–M) as a novel biomaterial for biomedical applications. The wound-dressing G–CH–M biopolymer was synthesized and characterized. The blood haemolysis, anti-inflammatory, antioxidant, and antibacterial activities of the biopolymer were investigated against Gram-positive (*Staphylococcus aureus*) and Gram-negative (*Escherichia coli*) bacterial isolates. Our results showed that *S. aureus* swarming motility was drastically affected. However, the biopolymer had no significant effect on the swarming motility of *E. coli*. In addition, the biopolymer showed high antibacterial capacities, especially against *S. aureus*. Plasmid DNA was observed to be effectively protected from oxidative stresses by the biopolymer. Furthermore, the biopolymer exhibited greatly suppressed haemolysis (lower than 2%), notwithstanding the elevated concentration of 50 mg/mL. These results indicated that this novel biopolymer formulation could be further developed for wound care and contamination prevention.

## 1. Introduction

Nowadays, the advanced development of materials based on biocompatibility and biodegradability is a long-term target for biomedical and bioengineering applications. The physicochemical characteristics of polymers have attracted tremendous attention in order to explore their potential use as natural and harmless materials [1].

Gelatin is considered to be one of the most versatile biopolymers among natural polymers. It has a unique set of biological properties, including its safe nature, biocompatibility, biodegradability, and absence of toxicity [2]. Thus, it is an attractive material in the drug, food, and tissue engineering sectors. The gelatin presently used in tissue engineering products is mainly of porcine or bovine origin. Nevertheless, the use of these gelatins tends to be more limited due to socio-cultural and disease-associated problems [3]. Therefore, the need for alternative sources of bovine and porcine gelatin has greatly increased. Recently, camel skin gelatin has been successfully isolated and characterized. It has been shown that camel skin gelatin has excellent physicochemical characteristics due to its mammalian source [4]. Chitosan is a naturally occurring polymer derived by the process of alkaline hydrolysis of chitin, a common chemical present in the shells of crustaceans and the outer coverings of insects. Chitosan has garnered significant interest for its favourable characteristics, such as its compatibility with living creatures, ability to be decomposed by natural processes, and lack of harmful or toxic effects [5]. As a result, it has found extensive use in the fields of food, drugs, and biomedicine. In many medical applications, both chitosan and gelatin have to be crosslinked in order to improve their physicochemical properties. Chitosan films have been evaluated for their therapeutic potential in wound dressings and have also been explored as scaffolds in tissue engineering [6,7]. In addition, chitosan exhibits effective antimicrobial properties against a wide range of microorganisms, including Gram-positive and Gram-negative bacteria, as well as filamentous fungi and yeast [8].

Recently, the incorporation of therapeutic agents into biomaterials has represented a novel approach to controlling inflammation, preventing infection, and stimulating tissue regeneration [9,10]. *Moringa oleifera* (*M. oleifera*) is a member of the Moringaceae family and was called “the miracle tree” or “tree of life” in old times because of its useful effects in curing various diseases. A wide variety of therapeutic virtues have been attributed to its roots, bark, leaves, flowers, fruits, and seeds. Phytochemical analyses have shown that the leaves of *M. oleifera* contain large amounts of protein, vitamins, β-carotene, flavonoids, and minerals. In fact, *M. oleifera* is widely utilised as a folk remedy to treat diabetes, malnutrition, anaemia, arthritis, and ulcers [11]. Several studies have shown that the leaf extract increases human dermal fibroblast proliferation, leading to faster wound healing [12,13]. Further, its antibacterial, antihypertensive, anti-inflammatory, antifungal, immunomodulatory, cardioprotective, and hepatoprotective activities have also been documented [14,15]. A few studies regarding the development of formulations loaded with *Moringa* extract intended for wound treatment as scaffolds, films, and hydrogels have also been recently reported [16,17,18].

This investigation is the first one to the best of our knowledge regarding the use of camel-skin-gelatin–chitosan–moringa as a novel biomaterial for biomedical applications. In the current study, the potential of using this film formulation as a functional biomaterial for pharmaceuticals was investigated. The antioxidant, blood haemolysis, and anti-inflammatory properties as well as the antibacterial effects of the film formulation were analysed.

## 2. Results

### 2.1. Rheological Properties

The G–CH–M biopolymer exhibited high porosity (79.4%) and low density (0.07 g/cm^3^). Another important parameter is the swelling capacity of biopolymer which serves as a crucial indicator for assessing biopolymers in tissue engineering applications. The swelling property of the G–CH–M biopolymer showed a swelling degree of 83.7% in phosphate buffer.

### 2.2. Antibacterial Activities

Agar well diffusion results showed that the G–CH–M biopolymer exhibited a potent inhibitory effect against tested bacteria (Figure 1). Increasing concentrations of the G–CH–M biopolymer resulted in a wider zone of inhibition. Wells 1 and 2 tested the G–CH–M biopolymer at concentrations of 15 and 20 mg/mL; the largest inhibition zones were observed at those concentrations (16 and 14 mm for *S*. *aureus* and 12 mm for *E*. *coli*). The overall results confirmed that the zone of inhibition of the biopolymer was greater in *S. aureus* than in *E. coli.* Furthermore, the antibacterial activity using the liquid medium microdilution assay exhibited a significant increase in inhibition activity against *S. aureus* after 24 h of incubation, while the biopolymer demonstrated a moderate antibacterial activity against *E. coli* (*p* < 0.001) (Figure 2).

### 2.3. Anti-Biofilm Activity

The anti-biofilm effects of the biopolymer determined using the crystal violet method are presented in Figure 3. The current findings show that biofilm enriched with moringa extract seemed to inhibit the initial binding of *E. coli* to the polystyrene plane. In fact, inhibition of 11.04% biofilm formation by *E. coli* in the presence of the G–CH–M biopolymer was observed. However, the biopolymer had moderate effect on the formation of biofilm by *S. aureus* after incubation for 24 h.

### 2.4. Modulation of Bacteria Motility

Figure 4 displays the analysis of the motilities of *S*. *aureus* and *E*. *coli* following incubation with the biopolymer. The motility of *S. aureus* was significantly reduced by the biopolymer. The treatment with the biopolymer showed a strong effect, as *S. aureus* showed a reduced swarm motility area. In contrast, the treatment had no discernible effect on the motility of the *E*. *coli* strain in comparison to the control group.

### 2.5. Antioxidant Activities

#### 2.5.1. DPPH Assay

The DPPH assay is often adopted to evaluate the total antioxidant power of various plants. The result of the DPPH free-radical-scavenging ability of the biopolymer is shown in Figure 5 and compared with ascorbic acid as a control standard. As can be seen from Figure 5, the DPPH radical scavenging increased from 60% to 95% when the concentration of the biopolymer increased from 0.125 to 2 mg/mL. The results indicated that the biopolymer exhibited potential DPPH radical-scavenging activity.

#### 2.5.2. Ferrous Chelating Activity

As shown in Figure 6, the metal chelating activity of the biopolymer increased with increasing concentrations used in the experiment. The biopolymer exhibited its maximum ferrous chelating activity at a concentration of 2 mg/mL.

### 2.6. DNA Damage Protective Effect Assay

The efficiency of the biopolymer in preventing oxidative damage to DNA is shown in Figure 7. The damage to plasmid DNA produces an open circular form and, further, a linear form of DNA. Therefore, the formation of circular and linear forms of DNA is indicative of single strand breaks and double strand breaks, respectively. The plasmid DNA was mainly of the supercoiled form (Form I) in the absence of Fe^2+^ and H_2_O_2_ (Figure 7, lane 1). After the addition of Fenton’s reaction mixture, the supercoiled form of DNA was converted into the circular and linear forms (Figure 7, lane 2), which indicate that hydroxy radicals generated from iron-mediated decomposition of H_2_O_2_ generated both single strand and double strand DNA breaks.

Our findings unequivocally demonstrate the safeguarding effect that the biopolymer provides against strand disruption caused by hydroxyl radicals in a Fenton’s reaction mixture (Figure 7, Lane 3).

### 2.7. Anti-Inflammatory Activity

Figure 8 displays the varying concentrations of the G–CH–M biopolymer and its corresponding in vitro anti-inflammatory efficacy. Increasing the concentration of the sample resulted in higher inhibition percentage for bovine serum albumin (BSA) denaturation. *M*. *oleifera* is extensively utilised in traditional medicine for the treatment of inflammatory conditions.

### 2.8. Blood Haemolysis Test

A haemolysis test was employed to determine the erythrocyte compatibility of the G–CH–M biopolymer. The haemolysis rates of the biopolymer with varied concentrations are shown in Figure 9. As seen in Figure 9, the biopolymer showed good biocompatibility with rabbit blood. According to the ASTM international standard practice for assessment haemolytic properties of materials [19] the haemolysis rate of blood contacting materials must be lower than 5%. It was observed that haemolysis rates of the biopolymer with different concentrations were all below 2%, showing a significant difference as compared with the control. The obtained results are in agreement with previous studies that reported that the haemolysis degrees of chitosan- and gelatin-based membranes exhibited excellent hemocompatibility.

## 3. Discussion

Infection of a wound with pathogens delays the healing process and causes severe wound dehydration. Therefore, it is important to evaluate the applicability of the developed wound dressing membranes for potential use in the healthcare sector. Our result demonstrates the antibacterial activity of the fabricated biopolymer against *S. aureus* and *E. coli*. A significant increase in inhibition activity was observed against *S. aureus* after 24 h of incubation. While the biopolymer demonstrates moderate antibacterial activity against *E. coli*, the observed outcome may be ascribed to the lipopolysaccharide’s (LPS) putative protective effect against Gram-negative bacteria. These results are similar to the observations of Thaya et al. [20] and Divya et al. [21], who demonstrated that a greater reduction in bacterial growth was observed after treatment with a biopolymer enriched with gelatin-coated zinc oxide nanoparticles or alginate. Furthermore, previous studies reported that both chitosan and chitosan–gelatin membrane-forming solutions had great inhibitory effects on *E. coli* and *L. monocytogenes* [8]. Similarly, Bukar et al. [22] confirmed that the moringa leaf aqueous extract had antibacterial activity against *S. aureus*, *E. coli*, and *S. typhimurium*. Therefore, integrating moringa leaf extract into wound dressing applications may present an intriguing prospect. This phenomenon can be attributed to the ongoing significance of natural products and organic by-product forms in the formulation of medications [23]. Consequently, the boundless array of potential therapeutic leads presented by biological diversity is a direct consequence.

In addition, the motility of *S. aureus* was significantly reduced, while that of *E*. *coli* was not affected. The difference in motility reduction observed between *S. aureus* and *E. coli* may be explained by the fact that these bacteria have different cell wall compositions and structures. *S. aureus* is a Gram-positive bacterium with a thick peptidoglycan layer, while *E. coli* is Gram-negative with a thinner peptidoglycan layer surrounded by an outer membrane. The differences in cell wall architecture could influence how they respond to motility-reducing agents (biopolymer in our case). Moreover, *E. coli* primarily relies on flagellar motility, which involves the rotation of flagella for movement. In contrast, *S. aureus* typically lacks flagella and instead utilizes alternative mechanisms such as surface-associated motility (e.g., twitching or gliding) or biochemical gradients (e.g., chemotaxis) for movement. These diverse motility strategies may respond differently to motility-reducing agents. In addition, formulated biopolymers act differently in gene expression profiles related to motility between *S. aureus* and *E. coli*; the motility inhibitors’ targets could contribute to the observed contrast in motility reduction.

It has been reported that bacterial motility is associated with biofilm development and antibiotic resistance [24,25]. Importantly, the current study shows that inhibition of bacteria swarming was associated with the *S*. *aureus* anti-biofilm activity of biopolymer. However, the motility assay conducted in plates containing the G–CH–M polymer revealed no impact on the ability of *E*. *coli* to swarm, unlike the observations with *S*. *aureus*.

These results suggest that the inhibition of *E*. *coli* biofilm by the formulated biopolymer did not lead to a decrease in swarm motility, as anticipated. This implies that, in the case of this bacteria, biofilm formation may be influenced by factors other than bacterial motility.

The current study reported the anti-biofilm activities of the G–CH–M biopolymer, where the inhibition of the initial attachment of *E. coli* to the polystyrene surface was observed. Biofilms are associated with growth and pathogenicity. Several studies have reported that biopolymers can inhibit biofilm formation by pathogenic bacteria [20,26]. As mentioned above, moringa contains a variety of bioactive compounds with antimicrobial properties that affect the growth, survival, and pathogenicity of different pathogens, which explains the reduction in biofilm formation when the strain was exposed to the biopolymer.

This study also demonstrated that this biopolymer base has antioxidant activities, which could be attributed to the presence of certain antioxidant compounds such as kaempferol, quercetin, and other polyphenolic molecules released from the moringa leaf extract [27]. Moringa is widely recognised for its exceptional antioxidant properties and its ability to effectively eliminate reactive compounds. Polyphenols exert several biological roles through their antioxidant activity. Moringa contains phenolic antioxidant chemicals that can effectively shield pHEN4 plasmid DNA from harm caused by Fenton’s reagents. This protection is achieved through a mechanism involving the transfer of electrons or hydrogen atoms, as previously reported [28].

The present study indicated that the G–CH–M biopolymer has good biocompatibility, and this is in agreement with previous studies that reported that the haemolysis degrees of chitosan- and gelatin-based membranes exhibit excellent hemocompatibility. Combining the results obtained from antioxidant activities and biocompatibility tests, the G–CH–M biopolymer might be said to prevent the inflammation reaction. Our results support the traditional use of *M. oleifera* to treat inflammatory disorders [29,30].

## 4. Materials and Methods

### 4.1. Chemical and Reagent

Sodium hydroxide (NaOH), citric acid (CH_2_COOH_2_), chitosan powder (90% DD, degree of deacetylation), ethanol (CH_3_CH_2_OH), 2,2-diphenyl-1-picryl-hydrazyl-hydrate (DPPH), ethylene diamine-tetraacetic acid (EDTA), bovine serum albumin (BSA), agarose, iron (III) chloride (FeCl_3_), hydrogen peroxide solution (H_2_O_2_), and L-ascorbic acid were purchased from Sigma-Aldrich (St. Louis, MO, USA), and LB broth, Petri dish, and 96-well polystyrene plates from (Thermo Scientific™. Waltham, MA, USA) Moringa was purchased from the local supermarket.

### 4.2. Extraction of Gelatin

The gelatin was extracted from camels’ skin as described previously [31]. Briefly, the camel skins of *Camelus dromedarius* were randomly collected from the slaughterhouse. The skins were washed thoroughly with distilled water and cut into small pieces. At room temperature, camel skin was immersed in 0.5 M NaOH with a skin/solution ratio of 1:5 (*w*/*v*) for 3 days to eliminate non-collagenous proteins. After thorough washing, the pre-treated skins were soaked in 0.1 M citric acid with a ratio of 1:5 (*w*/*v*) for 1 h. The samples were washed again with distilled water until the pH was neutral. The final step was carried out with distilled water at a ratio of 1:5 *w/v* at 50 °C for 6 h. The supernatant was passed through Whatman No. 4 filter paper, lyophilized, and subjected to analyses.

### 4.3. Polymer (G–CH–M) Construction

The polymer G–CH–M was prepared as described previously by Rahman et al., 2013 with some modification [32]. Briefly, the chitosan film forming solution (1% *w*/*w*) was prepared by dissolving the chitosan powder (90% DD, degree of deacetylation) in a 1% *w*/*w* solution of acetic acid and magnetically stirred for 8 h at room temperature until it completely dissolved. The gelatin derived from camel skin was solubilized in water (4%) and subjected to heating at 60 °C for a duration of 5–10 min. Prepared solutions were homogenized at a ratio of 1:1 for 30 min in order to assure their homogenization. Moringa solution was prepared by dissolving 1 g of dried leaves in distilled water and kept overnight at 4 °C. The resulting mixture was filtered through Whatman filter paper no. 1, and then, 1% (*w*/*w* based on the gelatin content) of extract was added to the mixture.

The gelatin/chitosan/moringa solution was lyophilized for 48 h after being frozen at −80 °C for 8 h. Finally, the dry films were exposed to UV light for 30 min for sterilization and subjected to biological analysis [33].

### 4.4. Rheological Properties

#### 4.4.1. Density

The density of the G–CH–M polymer was determined by measuring the dry weight (W), radius (R), and Height (H) of the polymer [34].

The density (p) of the polymer was calculated as follows:Density (p) = W/π × r^2^ × H

#### 4.4.2. Porosity

The porosity of the G–CH–M polymer was evaluated as described previously by Rajasree et al., 2020 [34]. Briefly, weighed polymer (Wd) was immersed in absolute ethanol and placed under reduced pressure in a dessicator for 15 min. After that, the polymer was removed and weighed (Ww) again. The porosity (%) was calculated by using the formula,
Porosity (%) = (Vb − Vl)/(Va − Vb)
where Vb and Va represent the volume of ethanol before and after immersion of the polymer, respectively. Vl represents the volume of ethanol after the removal of the sample from the liquid.

#### 4.4.3. Swelling Capacity

The swelling ratio of scaffold was measured by the conventional gravimetric method [35]. The dry weight of scaffold was measured before immersing in 0.05 M phosphate-buffered saline (PBS) pH 7.4 at 37 °C, and excess surface phosphate-buffered saline was blotted out with absorbent paper. The wet weight (Ws) of the scaffold was measured after incubation for 24 h at 37 °C. The percentage swelling of scaffolds were calculated by the formula,
WS (%) = [(W1 − W0)/W0] × 100
where W1 is the weight of the swollen polymer at equilibrium, W0 is the weight of the dry polymer.

### 4.5. Antibacterial Activity of G–CH–M Biopolymer

#### 4.5.1. Agar Diffusion Assay

The antibacterial activity of the G–CH–M biopolymer against both Gram-positive *Staphylococcus aureus* (*S. aureus* ATCC 12600) and Gram-negative *Escherichia coli* (*E. coli* ATCC 9637) was evaluated by the agar diffusion method [32]. The bacteria were spread onto the LB agar Petri dish with a 10^6^ CFU/mL density. Wells measuring 6 mm in diameter in the LB agar were filled with 30 μL of the prepared compound in comparison with ampicillin as a positive control. After that, the Petri dishes were incubated for 24 h at 37 °C. After incubation, the diameter of the zone of inhibition was measured. For each bacterial species, experiments were performed in triplicate assays.

#### 4.5.2. Liquid Medium Microdilution Assay

The antibacterial efficacy of the sterilised biopolymer was further assessed utilising microdilution techniques [36]. Briefly, 100 µL of bacterial suspensions (OD 0.1–0.2 at 600 nm) was inoculated into the wells of 96-well microtiter plates in the presence of 100 µL of the disinfected G–CH–M biopolymer with different final concentrations of (1 mg/mL, 2 mg/mL, 5 mg/mL, 10 mg/mL, 15 mg/mL, and 20 mg/mL) and incubated at 37 °C for 12 h, 18 h, and 24 h. The well containing only inoculum was employed as positive control. At each time, the absorbance value of the sample was determined at 600 nm. The bacterial inhibition percentage was determined as follows:bacterial inhibition (%) = Ic − Is/Ic × 100,
where Ic and Is are the absorbance values of the control inoculum and the inoculum containing biopolymers at predetermined times, respectively.

### 4.6. Crystal Violet Biofilm Assay

A static biofilm formation assay was performed in 96-well polystyrene plates as described previously, with some modifications [37]. In brief, *E. coli* and *S. aureus* were cultured in LB and thinned out using fresh LB to an OD600 of 0.02. A mixture of 200 μL of the bacterial culture and 200 μL of LB broth was combined and incubated at 37 °C for 24 h (stationary phase) without agitation. Plates were rinsed three times with phosphate-buffered saline (PBS) the following day after LB medium was discarded. Following this, bacterial biofilms were observed by staining for 15 min at ambient temperature with 125 μL of 0.1% crystal violet. Plates were subsequently rinsed, and the quantity of biofilm, which had been dissolved in 100 μL of 95% ethanol, was determined through OD measurement at 570 nm.

### 4.7. Motility Assay

Motility assays against *E. coli* and *S. aureus* were performed using 0.5% agar LB plates containing 0.8% glucose, as previously described [38]. Both *E. coli* and *S. aureus* were grown to an OD 600 of 1.0, and about 100 μL of cultures were placed in motility plates using a sterilized pipette tip. The samples at a concentration of 15 mg/mL were added to these bacteria and then incubated at 37 °C under microaerobic conditions for 18 h. Each experiment was performed using at least three independent replicates.

### 4.8. Antioxidant Activities

#### 4.8.1. DPPH Radical-Scavenging Activity

The scavenging of 2,2-diphenyl-1-picryl-hydrazyl-hydrate (DPPH) assays was performed as described previously [39]. A 100 µL volume of sample solutions with varying concentrations (ranging from 0.25 to 10 mg/mL) was combined with 100 µL of ethanol and a DPPH solution (0.002%). The reaction mixture was thereafter placed in a dark environment and allowed to incubate for a duration of 60 min. The measurement of absorbance was conducted at a wavelength of 517 nm utilising a 96-well microplate reader. Ascorbic acid served as the positive control. The DPPH radical-scavenging activity was assessed by employing the subsequent formula:DPPH radical − scavenging activity (%) =Doc − Dos/Doc × 100,
where Doc and Dos are the absorbances of control and sample, respectively. Each experiment was conducted in triplicate.

#### 4.8.2. Ferrous Chelating Activity

The ferrous chelating assay was conducted according to [40]. At ambient temperature, the mixtures were incubated for five minutes. Following this, 10 µL of a 5 mM ferrozine solution was added. The absorbance of each sample mixture was determined at 700 nm using a microplate reader following a 10 min reaction time. Ethylene diamine-tetraacetic acid (EDTA) at a concentration of 1 mg/mL was used as a positive control. The test was carried out in triplicate. The chelating activity was determined using the same equation for DPPH activity.

### 4.9. DNA Damage Protective Effect Assay

The efficacy of the biopolymer in safeguarding supercoiled pHEN4 plasmid DNA against H_2_O_2_ was assessed using the DNA nicking assay, adhering to a marginally altered iteration of the preceding account [41]. Concisely, the plasmid DNA of the supercoiled pHEN4 chromosome was isolated from JM109 *E*. *coli* bacteria utilising the PureLink™ Quick Plasmid Miniprep Kit, which is commercially available from Invitrogen. At room temperature, 0.5 µg of the purified plasmid DNA was incubated with 10 µL of sample (15 mg/mL). Following the addition of Fenton’s reagents (30 mM H_2_O_2_, 50 µM ascorbic acid, and 80 µM FeCl_3_), the mixture was incubated at 37 °C for a duration of 5 min. After incubation, the protective activity of various samples against DNA damage induced by hydroxyl radicals was evaluated using a 1% (*w*/*v*) agarose gel, followed by ethidium bromide staining. Later, photographs of the composites were taken using UV light. As a control, untreated DNA was utilised to assess the DNA strand fractures.

### 4.10. Blood Haemolysis

The haemolysis tests were used to evaluate the blood compatibility between the biopolymer and erythrocytes [19]. Eight hundred μL of the G–CH–M biopolymer at different concentrations (1 mg/mL, 3 mg/mL, 6 mg/mL, 12 mg/mL, 25 mg/mL, and 50 mg/mL) was added to 200 μL of fresh rabbit whole blood (whole blood/normal saline = 8:10). Following an hour of incubation at 37 °C, the samples were centrifuged at 4000 rpm for 5 min. The absorbance of the supernatant was measured at 545 nm. The experiments were run in triplicate. The haemolysis rate (HR) was calculated according to the following equation:HR = [(AS − AN)/(AP − AN)] × 100%,
where AS, AP, and AN present the absorbance of the biopolymer at different concentrations, the positive control (DI water), and the negative control (normal saline), respectively.

### 4.11. Anti-Inflammatory Activity

A protein denaturation technique was employed in accordance with the protocol outlined previously with slight modifications [33] to assess the anti-inflammatory activity of the G–CH–M biopolymer. Briefly, 1 mL of the biopolymer in different concentrations (100, 200, 500, and 1000 μg/mL) was added to 1 mL of PBS (pH 6.4) and 1 mL of bovine serum albumin solution (5%). For protein denaturation, samples were then incubated in a water bath at 72 °C for 10 min and cooled for 30 min at 25 °C. The turbidity of the solutions (level of protein precipitation) was measured at 660 nm. The experiments were conducted in duplicate, and the mean absorbance values were recorded. Negative (without samples) controls were assayed in a similar manner.

### 4.12. Statistical Analysis

All experiments were conducted in triplicate, and the results were presented as mean ± SD of triplicate measurements. The experimental data underwent analysis using ANOVA test, where a *p*-value of <0.05 was deemed statistically significant.

## 5. Conclusions

The biopolymer, formulated with camel-gelatin–chitosan and enriched with moringa, exhibits promising properties such as high antioxidant and anti-inflammatory activities, along with good rheological properties. These attributes are essential for its potential application in tissue engineering. Moreover, the study demonstrates that the combination of moringa, camel gelatin, and chitosan significantly affects the motility and biofilm formation of bacteria, particularly Gram-positive bacteria. Consequently, this newly developed biopolymer holds great potential for diverse applications in tissue engineering, particularly in wound dressings. Further investigations, including in vitro and in vivo studies using skin wound models, are warranted to validate its efficacy and safety.

## Figures and Tables

**Figure 1 pharmaceuticals-17-00545-f001:**
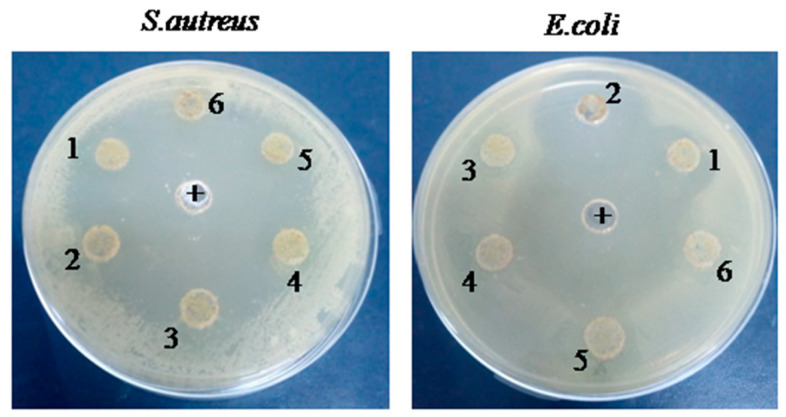
The antibacterial activity evaluation of G–CH–M biopolymer with different concentrations (1 to 6: 20, 15, 10, 5, 2, and 1 mg/mL) against *E. coli* and *S. aureus* using agar diffusion in comparison with ampicillin as a positive control (+).

**Figure 2 pharmaceuticals-17-00545-f002:**
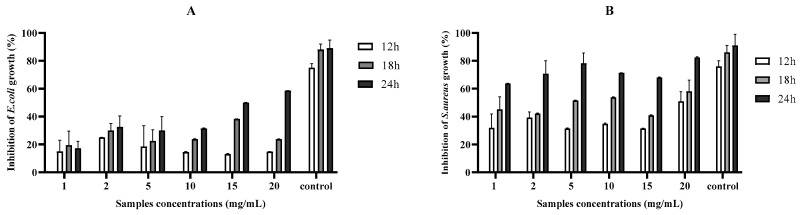
The antibacterial activity of G–CH–M biopolymer at different concentrations against (**A**): *E. coli* and (**B**): *S*. *aureus* using the liquid medium microdilution method. Error bars represent standard deviation of triplicates.

**Figure 3 pharmaceuticals-17-00545-f003:**
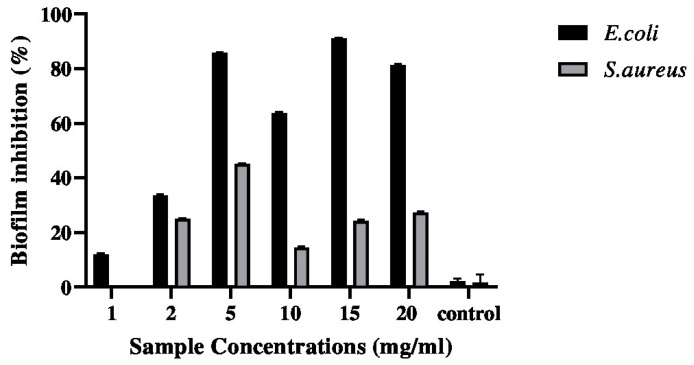
Effects of G–CH–M biopolymer on biofilm formation of *S. aureus* and *E. coli* after 24 h of incubation. Error bars represent standard deviation of triplicates.

**Figure 4 pharmaceuticals-17-00545-f004:**
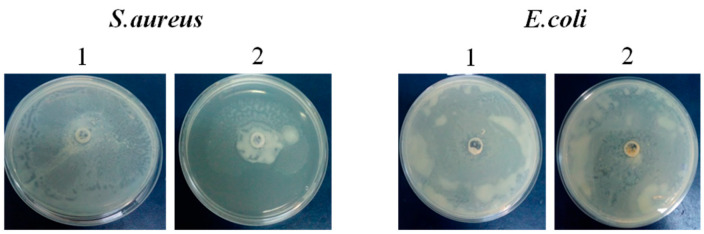
The motility of *Staphylococcus aureus* and *Escherichia coli* ante (1) and post (2) biopolymer treatment for 18 h.

**Figure 5 pharmaceuticals-17-00545-f005:**
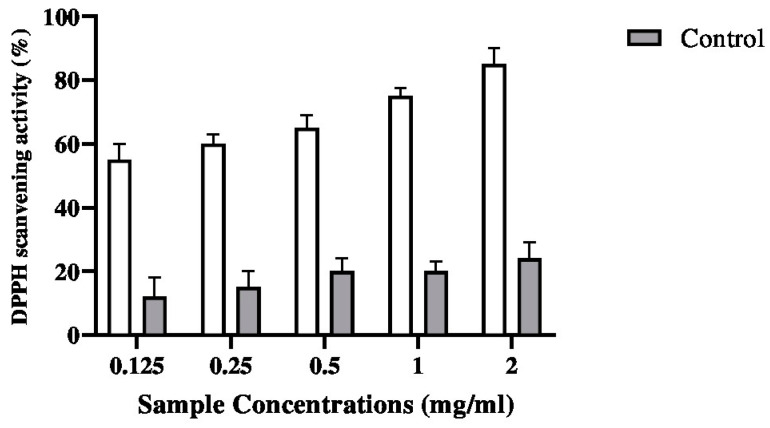
DPPH scavenging activity of G–CH–M biopolymer at different concentrations in comparison with biopolymer without moringa extract (control). Error bars represent standard deviation of triplicates.

**Figure 6 pharmaceuticals-17-00545-f006:**
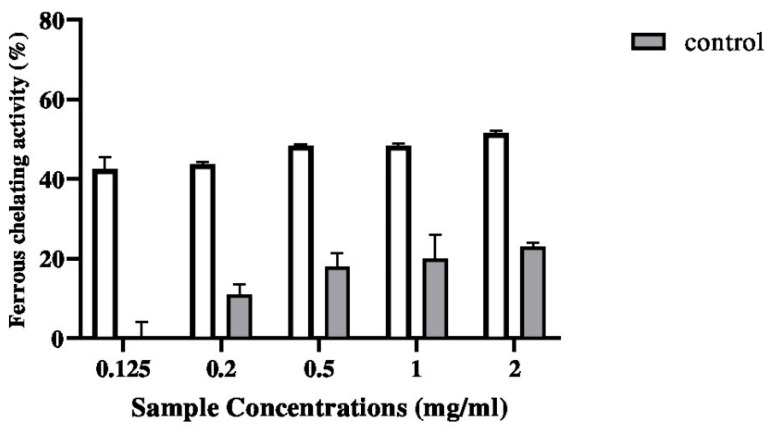
Ferrous chelating activity of G–CH–M biopolymer at different concentrations in comparison with biopolymer without moringa extract (control). Error bars represent standard deviation of triplicates.

**Figure 7 pharmaceuticals-17-00545-f007:**
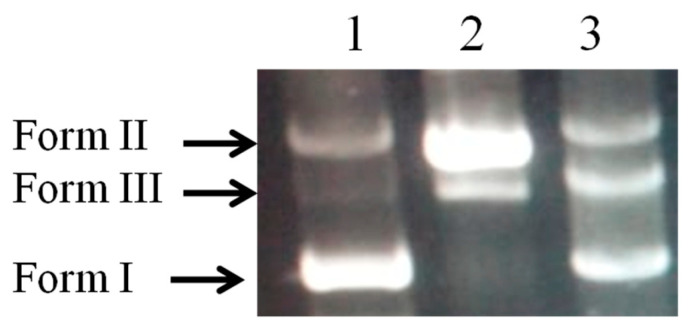
Protective effect of biopolymer on plasmid DNA nicking caused by hydroxyl radicals. Lane (1) native pHEN4 DNA; Lane (2) DNA plasmid + Fenton’s reagent; Lane (3) native pHEN4 DNA+ biopolymer Fenton’s reagent. Where form I, form II, and form III designed supercoiled form, nicked circular form, and linear form of plasmid, respectively.

**Figure 8 pharmaceuticals-17-00545-f008:**
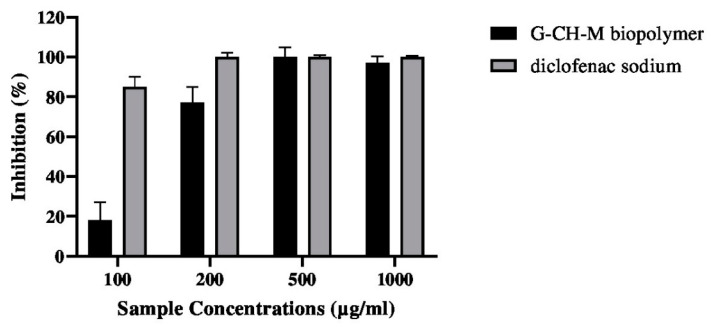
Effect of G–CH–M biopolymer on BSA denaturation. Error bars represent standard deviation of triplicates.

**Figure 9 pharmaceuticals-17-00545-f009:**
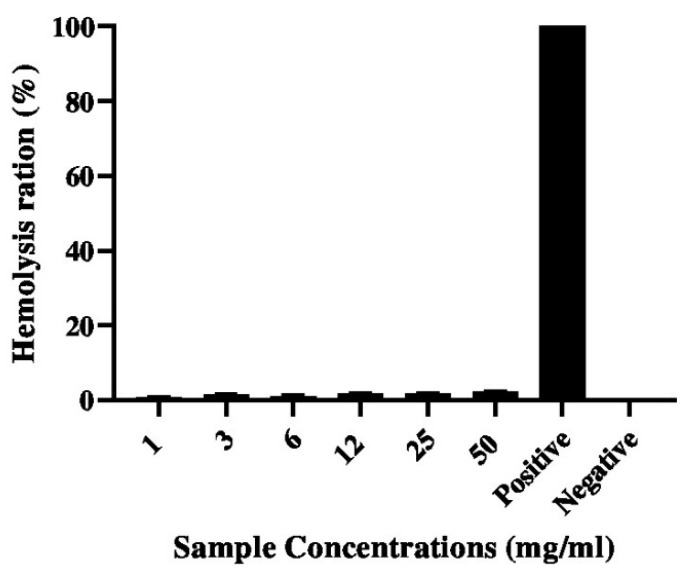
Haemolysis ratios of RBCs treated with different concentrations of G–CH–M biopolymer at 37 °C for 3 h.

## Data Availability

The data presented in this study are available on request from the corresponding author.

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
