# Peer review of "Antibacterial, Anti-Biofilm, and Anti-Inflammatory Properties of Gelatin–Chitosan–Moringa-Biopolymer-Based Wound Dressings towards Staphylococcus aureus and Escherichia coli"

_pharmaceuticals, 2024, doi:10.3390/ph17050545_

Round 1

Reviewer 1 Report

Comments and Suggestions for Authors

Faraz et al provide an interesting approach to infuse moringa leaf extract into biopolymers to fabricate a wound dressing. Wound healing is indeed one of the challenges for pharmaceutical research due to biofilm formation as well as toxicity concerns. The authors have provided a good number of methodologies to address different aspects of their suggested G-CH-M wound dressing. Unfortunately, the paper falls short in many respects. Therefore, we advise against publishing this manuscript in Pharmaceuticals journal. The comments and reasons are enlisted below:

·         A very important control is missing in every setting which is the biopolymer without moringa extract.

·         The authors do not provide any reasoning why they chose the ratios and concentrations of the composition of their polymer.

·         Section 2.1: The authors should check reproducibility by comparing batch-to-batch extracted gelatin as this was extracted by themselves from randomly chosen animals. So reproducibility concerns shall arise.

·         Section 2.2: The amount of moringa solution is not stated in the methods as well as the mixing ratios to make the G-CH-M composition.

·         Section 2.5: What concentration was used for the motility assay? This information is missing.

·         Section 2.6.2: What concentration of EDTA was used for the positive control?

·         Section 2.7: What concentration was used for the DNA damage assay?

·         Section 2.7: The authors used ascorbic acid as a claimed reagent for Fenton’s reaction. Ascorbic acid is a good radical scavenger and antioxidant, so it does the opposite of the intended use.

·         Section 2.8: Biocompatibility should not only be done with hemolysis testing but also be investigated by cytotoxicity assays, preferably fibroblasts or skin cell line.

·         Section 2.8: Distilled water should not be used as a positive control for hemolysis. Most literature takes 1% TritonX as a standard or distilled water with freeze-thaw cycles as an alternative.

·         Section 2.9: Reference 24 discusses protein denaturation as a “screening” tool for anti-inflammatory drugs. Therefore, it can not be used for quantifying nor confirming the anti-inflammatory effect of the biopolymer.

·         Section 3.1 (figure 1): Only concentrations 15 and 20 mg/mL show some inhibition zone. These two concentrations are very high rendering them not applicable for any practical use.

·         Section 3.1 (figure 1): 15 mg/mL looks better than 20 mg/mL in terms of antimicrobial activity. Does the authors have any explanation for this?

·         Section 3.1 (figure 2): If 20 mg/mL can not inhibit bacterial growth, then simply the material is not active. 20 mg/mL of most molecules, even if not antimicrobial, will inhibit at such concentrations.

·         In figures 2, 4, and 6: The concentration do not follow a trend. In many cases, a lower concentration is more active than a higher concentrations. This raises concerns about the validity of this data, therefore, more biological replicates are needed.

·         Figures 3 and 4 are swapped

·         Section 3.3: Huge differences exist between figure 3 and reference 19, which the authors used as a source for their method and interpretation.

·         Section 3.4.1: The used concentrations in this assay (DPPH assay) is very high and not applicable.

·         Section 3.5: The authors probably mistyped “biofilm” instead of “biopolymer” in line 259 and line 265.

·         Something is wrong with the text in lines 294-296 and 320-321. They are not understandable.

·         In line: Not any “biopolymer” can inhibit biofilm formation

·         The authors do not offer any explanation why S. aureus motility was reduced and not E. coli, provided that their methods gets improved in this assay.

·         In the references, journal names should be italicized.

Comments on the Quality of English Language

The English can be improved

Reviewer 2 Report

Comments and Suggestions for Authors

The manuscript raises many interesting topics. The preliminary experiments are promising. However, the implementation of the summarized experiments often creates a feeling of lack. The choice of types of experiments cannot be said to be prudent. The experimental results do not provide a clear basis for the strong conclusions made. I recommend further in vitro determinations. Hemolysis assay as a biocompatibility test alone is not conclusive. I recommend MTT, LDH or equivalent tests on cell cultures (e.g. HaCaT).

Errors in experimental methods should also be corrected. The quality and origin of the materials used are often not specified clearly. Please correct! You mention that "The antibacterial efficacy of the sterilized biopolymer was further assessed utilizing microdilution techniques". This is not specified in detail. In addition, the impact of the sterilization process should be examined as well.

The manuscript mentions that "Protein denaturation technique was employed in accordance with the protocol outlined by Williams et al. to assess the anti-inflammatory activity of the G-CH-M biopolymer." I don't think that's the best solution for that. The level of some specific inflammatory mediator (or rather several) should be analyzed.

Figures are poorly edited. My key observations are:

Figure 2 - no standard deviation indication

Figure 3 -measurement sample number missing

Figures 5 and 6 - although the standard deviation is visible, its extent is not interpretable, and there is no reference to standard deviation in the Figure legend.

Figure 9 -no standard deviation is indicated. It is not clear whether any standard deviation was calculated from the results or exactly how many parallel measurements were made.

The conclusion is too short, yet too exaggerated in drawing conclusions from the results.

The author does not share any information about statistical analysis. The statistical methods used to evaluate the experiments shall be provided as well.

However the manuscript has real value, in its current form, it has a lot to improve! Please review and improve it carefully.

Reviewer 3 Report

Comments and Suggestions for Authors

The manuscript entitled "Antibacterial, anti-biofilm and anti-inflammatory properties of G-CH-M biopolymer-based wound dressings towards Staphylococcus aureus and Escherichia coli" deals to investigate the potential of gelatin-chitosan-moringa leaf extract as a novel biomaterial for biomedical applications. Overall, the manuscript is well written and organized. However, to improve the manuscript's quality a major revision is required.

Major Comments:

1.      Include the characterization of gelatin-chitosan-moringa dry films.

2.      Include thixotropic characteristics of the developed dry films.

3.      Check the figure numbers carefully and maintain sequence; Figure 9 was listed as before Figure 8.

The manuscript needs to be revised and needs major revision before considering it for further review or reconsideration.

Reviewer 4 Report

Comments and Suggestions for Authors

Dear author,

The paper entitled “Antibacterial, anti-biofilm and anti-inflammatory properties of G-CH-M biopolymer-based wound dressings towards Staphylococcus aureus and Escherichia coli” has been intensively reviewed and evaluated. Although the present study was considered an interesting study, some points need to be revised. Hereby, I would like to present my suggestions and revisions.

Revision_1: The introduction needs to be expanded and supported by more references. I recommend revising the whole part as it is insufficient. In particular, the part regarding chitosan has no citation. For example, there are papers in the literature where the antibacterial activity of chitosan is tested, which should be cited and commented on.

Revision_2: The materials paragraph is missing.

Revision_3: In almost all biological tests the control is missing, which needs to be added.

Revision_4: This paper lacks the significance of the experimental data. I suggest the authors calculate it, after testing the controls in each assay.

Revision_5: The authors did not perform any chemical-physical and/or morphological characterization tests of this new entity (G-CH-M). How can they be sure that chitosan has bound? At least one characterization is needed after synthesis to be certain that polymer synthesis has occurred. Attached is a paper from which you can choose and argue a characterization (doi: 10.3390/pharmaceutics14050942; doi.org/10.1002/app.26402)

Revision_6: In Figure 7 the forms I II III are not adequately described

Revision_7: Figure 8 needs to be improved because it is in bad resolution.

Revision_8: In some images, there are axis titles, in others, there are not. I suggest that the authors unify all images.

Revision_9: In Figure 9 there is no standard deviation. Did the authors take only one test? It is suggested that the authors do at least one triplicate for each test.

Reviewer 5 Report

Comments and Suggestions for Authors

Dear Author,

Your manuscript titled "Antibacterial, anti-biofilm, and anti-inflammatory properties of G-CH-M biopolymer-based wound dressings towards Staphylococcus aureus and Escherichia coli" has undergone a thorough review. Following a meticulous assessment of its originality and content alignment, it has been concluded that your manuscript is in accordance with the criteria established by the Pharmaceuticals journal. However, I regret to inform you that certain major issues have been identified in your prepared manuscript. The outlined issues are presented below as major requests. I anticipate your careful attention to these matters, providing responses that align with the specified requirements during the revision process.

The issues are as follows:

1. Please check the title (Line 3): Please avoid from using abbreviations in the title (It is kindly recommended that using the unabbreviated form of “G-CH-M”.)

2. Please check (L68): (as a recommendation) to prevent the confusion of meaning “antiulcer” could be changed to “ulcer”.

3. (First major concern): Please clarify the confusion according to the statement at L79. It was understood a thin film formulation was formed (a solid one) from the expression “…antibacterial effects of the film formulation…” Therefore, a physicochemical characterization must be performed (FTIR analysis to illuminate the excipient compatibility, Tensile strength, elongation at break, Young’s modulus measurement to demonstrate the mechanical characteristics..etc.). From the physicochemical perspective the the study could be considered as insufficient. Please support the study from this point of view.

4. (Second major concern): There were no information about the preparation of “moringa's extraction/solution process” Please indicate the M. oleifera addition and also there were a critical need to which major/stable phytocontent is responsible from the bioactivity (L322-325: This study also demonstrated that this biopolymer base has antioxidant activities and thus could be attributed to the presence of certain antioxidant compounds such as kaempferol, quercetin, and other polyphenolic molecules released from the moringa leaf extract [33].) – Please demonstrate the possible phytocontent by using a suitable analysis method.

5. The Figure orders has shifted. Please control the entire figure numbers. Moreover please check the figure caption “Figure 4. Effects of biopolymer on biofilm formation of S. aureus and E. coli. after how many hours of incubation?”.

6. (Third major concern): “…fresh rabbit whole blood (whole blood: normal saline = 8: 10).” (L177). The fresh biomaterials needs Ethical approval or Ethical committee waivers, please provide related ethical form.

7. Please check your references (REF20, 25-27, 29…etc.) (Instruction for authors section rules authors to use the abbreviated form of journal names.)

Respectfully,

Round 2

Reviewer 1 Report

Comments and Suggestions for Authors

Most of the previous comments were not taken into consideration (see highlighted below), hence the quality of the manuscript at this current state is still not suitable for publishing at Pharmaceuticals.

Previous review:

Faraz et al provide an interesting approach to infuse moringa leaf extract into biopolymers to fabricate a wound dressing. Wound healing is indeed one of the challenges for pharmaceutical research due to biofilm formation as well as toxicity concerns. The authors have provided a good number of methodologies to address different aspects of their suggested G-CH-M wound dressing. Unfortunately, the paper falls short in many of these aspects. Therefore, we advise against publishing this manuscript in Pharmaceuticals journal. The comments and reasons are enlisted below:

·         A very important control is missing in every setting which is the biopolymer without moringa extract.

·         The authors do not provide any reasoning why they chose the ratios and concentrations of the composition of their polymer.

·         Section 2.1: The authors should check reproducibility by comparing batch-to-batch extracted gelatin as this was from animal source so reproducibility concerns shall arise.

·         Section 2.2: The amount of moringa solution is not stated in the methods as well as the mixing ratios to make the G-CH-M composition.

·         Section 2.5: What concentration was used for the motility assay? This information is missing.

·         Section 2.6.2: What concentration of EDTA was used for the positive control?

·         Section 2.7: What concentration was used for the DNA damage assay?

·         Section 2.7: The authors used ascorbic acid as a claimed reagent for Fenton’s reaction. Ascorbic acid is a good radical scavenger and antioxidant so it does the opposite of the intended ues.

·         Section 2.8: Biocompatibility should also be investigated by cytotoxicity assays against a skin cell line.

·         Section 2.8: distilled water should not be used as a positive control for hemolysis. Most literature takes 1% TritonX as a  standard or distilled water with freeze-thaw cycles as an alternative.

·         Section 2.9: Reference 24 discusses protein denaturation as a “screening” tool for anti-inflammatory drugs. Therefore, it can not be used for quantifying nor confirming the anti-inflammatory effect of the biopolymer.

·         Section 3.1 (figure 1): Only concentrations 15 and 20 mg/mL show some inhibition zone. These two concentrations are very high rendering them not applicable for any practical use.

·         Section 3.1 (figure 1): 15 mg/mL looks better than 20 mg/mL in terms of antimicrobial activity. Does the authors have any explanation for this?

·         Section 3.1 (figure 2): If 20 mg/mL can not inhibit bacterial growth, then simply the material is not active. 20 mg/mL of most molecules, even if not antimicrobial, will inhibit at such concentrations.

·         In figures 2, 4, and 6: The concentration do not follow a trend. In many cases, a lower concentration is more active than a higher concentrations. This raises concerns about the validity of this data, therefore, more biological replicates are needed.

·         Figures 3 and 4 are swapped

·         Section 3.3: Huge differences exist between figure 3 and reference 19, which the authors used as a source for their method and interpretation.

·         Section 3.4.1: The used concentrations in this assay (DPPH assay) is very high and not applicable.

·         Section 3.5: The authors probably mistyped “biofilm” instead of “biopolymer” in line 259 and line 265.

·         Something is wrong with the text in lines 294-296 and 320-321. They are not understandable.

·         In line: Not any “biopolymer” can inhibit biofilm formation

·         The authors do not offer any explanation why S. aureus motility was reduced and not E. coli, provided that their methods gets improved in this assay.

·         In the references, journal names should be italicized.

Comments on the Quality of English Language

The language was improved compared to the intial submission, but not sufficiently

Author Response

Response: We thank the reviewer for the commitment and efforts to review our manuscript as well as for your contribution to improve the quality of our manuscript.

      A very important control is missing in every setting which is the biopolymer without moringa extract.

Response: Thank you for comments. A control test was added to each test

  • Section 2.1: The authors should check reproducibility by comparing batch-to-batch extracted gelatin as this was from animal source so reproducibility concerns shall arise.

Response: Thank you for comment. We agree with reviewer and understand the critical need for reproducibility. We have checked and compared with our previous protocol. therefore, we confirm that, this extract is reproducible

.

  • Section 2.7: The authors used ascorbic acid as a claimed reagent for Fenton’s reaction. Ascorbic acid is a good radical scavenger and antioxidant so it does the opposite of the intended ues.

Responses: Thank you for comment. We agree with reviewer that Ascorbic acid its has good antioxidant properties. in the current study we used polymers without moringa extract as control. and the Ascorbic acid as standard positive control. 

  • Section 2.8: Biocompatibility should also be investigated by cytotoxicity assays against a skin cell line.

Response: Thank you for the comment. We agree with reviewer that biocompatibility should also done using toxicity test in cells. Due to limited resources lack of cell culture in our lab. We have replaced the biocompatibility test to haemolysis test only.

  • Section 2.8: distilled water should not be used as a positive control for hemolysis. Most literature takes 1% TritonX as a  standard or distilled water with freeze-thaw cycles as an alternative.

Response: Thank you for comment and suggestion. We agree with the reviewer that 100% TritonX is optimal reagent can be use as control positive in haemolysis test in common literature book. However, the dH2O also found to produce 100% haemolysis, and used as control in some published work (https://doi.org/10.1016/S0142-9612(00)00163-0, and  https://doi.org/10.1002/jbm.b.33169) We used instead of triton since we don not have 100% Triton X.

  • Section 2.9: Reference 24 discusses protein denaturation as a “screening” tool for anti-inflammatory drugs. Therefore, it can not be used for quantifying nor confirming the anti-inflammatory effect of the biopolymer.

Response: Thank you for comment and suggestion. The reference 24 has replaced by Raoufi & Abdollahi (2023) who used protein denaturation test to evaluate the in vitro anti-inflammatory potential of formulated film.

  • Section 3.1 (figure 1): Only concentrations 15 and 20 mg/mL show some inhibition zone. These two concentrations are very high rendering them not applicable for any practical use

Response: Thank you for the comments. The concentration 15 and 20 mg/mL that only show inhibition in the bacteria of E.coli. In contract , all contraction tested was show inhibition zone toward S. aureus , suggested that the polymers efficacy depended on bacteria type . Therefore , the polymers was effective toward skin infection associated with S. aureus

  • Section 3.1 (figure 1): 15 mg/mL looks better than 20 mg/mL in terms of antimicrobial activity. Does the authors have any explanation for this?

Response: Thank you for the comments.  After, several replicated we observed that 15 mg/ml looks better than 20 . This is only in case of E.coli , since we reported that this bacteria is quite resistance to our tested polymers compare to S. aurues . We did not observed this phenomenon in case of S. aurues . Where the bacteria response better to higher concentration than lower. 

For the E.coli , we need to investigate more in our future experiments to understand the why its resistance , and why small dose is better .

  • Section 3.1 (figure 2): If 20 mg/mL can not inhibit bacterial growth, then simply the material is not active. 20 mg/mL of most molecules, even if not antimicrobial, will inhibit at such concentrations.

Response: Thank you for the comments.  All the concentration tested was inhibit the growth of S. aureus. In contrast, E.coli need higher concentration to inhibit the bacteria growth.

  • In figures 2, 4, and 6: The concentration do not follow a trend. In many cases, a lower concentration is more active than a higher concentrations. This raises concerns about the validity of this data, therefore, more biological replicates are needed.

Response: Thank you for comment. The test was repeated three times. The figures were amended accordingly

  • Section 3.3: Huge differences exist between figure 3 and reference 19, which the authors used as a source for their method and interpretation.

Response: Thank you for the comments. Reference 19 has been changed

  • Section 3.4.1: The used concentrations in this assay (DPPH assay) is very high and not applicable.

Response: Thank you for the comments. We have reduced the concentration from ( 0.5 – 8 mg/ml ) to (0.125 – 2 mg/ml)

  • The authors do not offer any explanation why S. aureus motility was reduced and not E. coli, provided that their methods gets improved in this assay.

Response: Thank you for the comments. We have explained in the differences in discussion section (Line 239-250).

Reviewer 2 Report

Comments and Suggestions for Authors

Dear Authors,

I accept your answers and thank you for the corrections you made in the manuscript. 

However the basic concept is well developed, the value of the manuscript is still not outsatnding. During the experimental implementation there are some incomplete and not very modern solutions. This observation had been made when I was criticizing the previous version. I accept the authors' explanation that they did not have opportunity to carry out the experiments that I suggested. Based on this, the manuscript has a low average scientific value. I recommend to consider the acceptance with this comment.

Author Response

Response to Reviewer 2- round 2 

Response:  We thank the reviewer for the commitment and efforts to review our manuscript as well as for your contribution to improve the quality of our manuscript. We have highlighted the limitation in conclusion section for further investigation in the future

Reviewer 3 Report

Comments and Suggestions for Authors

The authors have addressed all the raised comments, and the manuscript can be accepted in its current form.

Author Response

Response to Reviewer 3- round 2

Response:  We thank the reviewer for the commitment and efforts to review our manuscript as well as for your contribution to improve the quality of our manuscript. 

Reviewer 4 Report

Comments and Suggestions for Authors

I accept the manuscript in the present form

Author Response

Response to Reviewer 4-round 2

                   Response:  We thank the reviewer for the commitment and efforts to review our manuscript as well as for your contribution to improve the quality of our manuscript. 

Reviewer 5 Report

Comments and Suggestions for Authors

Dear Author,

Your revised manuscript has been thoroughly evaluated. As a result of the review, it has been determined that major deficiencies persist. Below are listed requests that need to be carefully reviewed and addressed. It is imperative that the requested actions be completed successfully.

1. (As a major request) The referencing format must be carefully followed (Such as Rheological characterization part: numbering reference style must be followed and also cited works must be inserted into the references section)

2. (As previously requested, – major revision must be fulfilled): Even if rheological characterization studies are evaluated as significant, the results of the FTIR study demonstrating physicochemical interactions are important for elucidating the interactions of formulation components. Since the FTIR device is commonly found in R&D laboratories, conducting this analysis is expected.

3. (As previously requested, – major revision must be fulfilled): Although the preparation of the moringa extraction/solution process was successfully carried out, the phytochemical content has not been demonstrated by the authors. (The authors could prove the existence of mentioned phytoactives)

Best regards

Author Response

Your revised manuscript has been thoroughly evaluated. As a result of the review, it has been determined that major deficiencies persist. Below are listed requests that need to be carefully reviewed and addressed. It is imperative that the requested actions be completed successfully.

Response : We thank the reviewer for the commitment and efforts to review our manuscript as well as for your contribution to improve the quality of our manuscript.

  1. (As a major request) The referencing format must be carefully followed (Such as Rheological characterization part: numbering reference style must be followed and also cited works must be inserted into the references section)

Response: Thank you for comments. We have corrected the referencing format as suggested

  1. (As previously requested, – major revision must be fulfilled): Even if rheological characterization studies are evaluated as significant, the results of the FTIR study demonstrating physicochemical interactions are important for elucidating the interactions of formulation components. Since the FTIR device is commonly found in R&D laboratories, conducting this analysis is expected.

Response: We totally understand the important of physicochemical characterization using FTIR. Unfortunately, the FTIR machine is not functioned in our lab at moment, and we do not have enough fund to purchase new one or repaired it.

  1. (As previously requested, – major revision must be fulfilled): Although the preparation of the moringa extraction/solution process was successfully carried out, the phytochemical content has not been demonstrated by the authors. (The authors could prove the existence of mentioned phytoactives)

Response: Thank you for comments. We totally agree with reviewer for the importance of performing phytochemical; analysis for moringa. Unfortunately, we are an able to carry out this experiment at moment, due to our finical and resource limit. We highlighted the bioactive contents for antioxidant based on the previous repost (ref 33).

Round 3

Reviewer 1 Report

Comments and Suggestions for Authors

Thanks to the authors for editing the manuscript as per the suggested comments. However, there are still some major comments on the manuscript:

·        For most MIC determination assays, the usual threshold for calling the concentration as inhibiting the bacteria is 80-95% meaning that the MIC for S. aureus and E. coli are around 20 mg/mL and >20 mg/mL. By any means, this is too high and not applicable by any means. I suggest the author to conduct a MIC experiment in accordance to the CLSI guidelines (5*10^5 CFU/mL inoculum) as well as subtracting the OD600 background of the well+medium.

·        In vitro cytotoxicity studies still needs to be conducted. Since you most of your system is extracted from animals or plants, it is very likely that there are toxins found in this extract, which could have a mechanism of toxicity other than the membrane damage.

·        It is not clear what control was added to the antibacterial activity assays. There should be a dilution series of moringa leaf extract only and the polymer without the leaf extract. It has to be proved that such combination is of any added value.

·        Rahaman et al 2013 in the journal of . Polymer International, 62(1), 79-86 should be mentioned and cited in the manuscript to justify choosing the concentrations and the ratios to the reader.

·        The motility reduction of S. aureus, but not E. coli needs an explanation. The provided explanation in line 231 is not enough. 

Comments on the Quality of English Language

N/A

Author Response

Thanks to the authors for editing the manuscript as per the suggested comments. However, there are still some major comments on the manuscript:

Response: We thank the reviewer for the commitment and efforts to review our manuscript as well as for your contribution to improve the quality of our manuscript.

  • For most MIC determination assays, the usual threshold for calling the concentration as inhibiting the bacteria is 80-95% meaning that the MIC for S. aureusand E. coli are around 20 mg/mL and >20 mg/mL. By any means, this is too high and not applicable by any means. I suggest the author to conduct a MIC experiment in accordance to the CLSI guidelines (5*10^5 CFU/mL inoculum) as well as subtracting the OD600 background of the well+medium.

Response: Thank you for comments.  Our study aimed to investigated the antibacterial activity of polymers using agar diffusion test methods. We used wide range of concentration (20, 15, 10, 5, 2, and 1 mg/ml) . All the concentration tested has some degree of bacteria inhibition. We understand the important of performing the MIC test to stablish the optimal concentration. Unfortunately, we are unable to perform the suggested test at the moment due to our resources limit and we don not have enough polymers materials at moments. We will carefully consider your suggestion for future research

  • In vitro cytotoxicity studies still needs to be conducted. Since you most of your system is extracted from animals or plants, it is very likely that there are toxins found in this extract, which could have a mechanism of toxicity other than the membrane damage.

Response: We acknowledge the importance of assessing the safety profile of our materials, especially when derived from natural sources like plants and animals.  Unfortunately, we are not able to perform the toxicity test at moment due to the lack of facilitates and resources. Therefore, we did the hemolysis test only. We will carefully consider your suggestion for future research endeavors and explore the feasibility of incorporating in vitro cytotoxicity assays to provide a more comprehensive assessment of safety.

Before incorporating these materials into our study, we conducted a comprehensive literature review to assess their safety profiles and any reported cytotoxic effects ( Line 49- 51) line ( 58- 62). Gelatin and chitosan are widely used biomaterials with established safety profiles in biomedical applications. They have been extensively studied for their biocompatibility and low cytotoxicity. Gelatin extracted from camel was fully characterised in our previous study and it contains up to 92% pure protein. It has been proved that gelatin lack of antigenicity or toxicity to cells (bello et al., 2020). chitosan was bought from sigma company. Also chotisan it is FDA approveduse in wound dressings and known for its biocompatibility and low cytotoxicity and non-antigenic .  Furthermore, our moringa extract was obtained through a carefully controlled extraction process to minimize the presence of any potentially harmful compounds.

  • It is not clear what control was added to the antibacterial activity assays. There should be a dilution series of moringa leaf extract only and the polymer without the leaf extract. It has to be proved that such combination is of any added value.

Response: the ampicillin was used as positive control for antibacterial testing used agar well diffusion methods. Both bacteria were knowing as sensitive to ampicillin.  We have mentioned in the text as you suggested For the dilution serious of moringa leaf extract and polymer without leaf extract. We decided to used this combination based prior literature described by Rahaman et al 2013, indicating the potential synergistic effects of combining natural extracts with polymer matrices for antimicrobial applications. Additionally, preliminary studies have suggested promising results with this particular combination in other contexts

  • Rahaman et al 2013 in the journal of . Polymer International, 62(1), 79-86 should be mentioned and cited in the manuscript to justify choosing the concentrations and the ratios to the reader.

Response: Thank you for comment and suggestion. We cited and mentioned the reference as you suggested.

  • The motility reduction of S. aureus, but not E. colineeds an explanation. The provided explanation in line 231 is not enough. 

Thank you for comments. More explanation is added to the text as suggested

Reviewer 5 Report

Comments and Suggestions for Authors

Dear Author,

It is evident from your explanations that certain requests could not be met due to limitations in your research resources. Taking this into consideration, I am inclined to provide a positive opinion regarding the acceptance of your manuscript for publication. However, I would like to remind you that in the previous version of your manuscript, I requested not to use abbreviations in the title. I kindly request that you consider this as you proceed, and with that in mind, I offer a positive opinion on the acceptance of your work.

Respectfully,

Author Response

We thank the reviewer for the commitment and efforts to review our manuscript as well as for your contribution to improve the quality of our manuscript.

The abbreviation has been removed from the manuscript title as you suggested